

# Excitation transfer in disordered spin chains with long-range exchange interactions

Nikolaos E. Palaiodimopoulos[1⋆], Maximilian Kiefer-Emmanouilidis[2,3], Gershon Kurizki[4] and David Petrosyan[1]

**1** Institute of Electronic Structure and Laser, FORTH, GR-70013 Heraklion, Greece
**2** Department of Physics and Research Center OPTIMAS,
University of Kaiserslautern-Landau, D-67663 Kaiserslautern, Germany
**3** Embedded Intelligence, German Research Centre for Artificial Intelligence,
D-67663 Kaiserslautern, Germany
**4** Department of Chemical and Biological Physics,
Weizmann Institute of Science, Rehovot 7610001, Israel

⋆ nikpalaio@iesl.forth.gr

## Abstract

We examine spin excitation or polarization transfer via spin chains with long-range exchange interactions in the presence of diagonal and off-diagonal disorder. To this end, we determine the mean localization length of the single-excitation eigenstates of the chain for various strengths of the disorder. We then identify the energy eigenstates of the system with large localization length and sufficient support at the chain boundaries that are suitable to transfer an excitation between the sender and receiver spins connected to the opposite ends of the chain. We quantify the performance of two transfer schemes involving weak static couplings of the sender and receiver spins to the chain, and time-dependent couplings realizing stimulated adiabatic passage of the excitation via the intermediate eigenstates of the chain which exhibits improved performance.

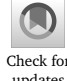

# 1 Introduction

Excitation or polarization transfer in interacting few- and many-body quantum systems plays a key role in many branches of science and technology, ranging from photosynthesis, where photon energy is transferred from a light-absorbing center to a reaction center via collections of near-resonant two-level systems (spins) [1], nuclear magnetic resonance of large molecules involving many interacting spins [2], or quantum state transfer in various spin chains realized, e.g., by dopants in solids [3–5], arrays of polar molecules [6,7], superconducting qubits [8], ions in traps [9,10] or Rydberg atoms in microtraps [11]. Whereas spin chains are commonly described in the nearest-neighbour approximation, experimentally relevant systems often possess long-range exchange interactions, or hopping, scaling with distance $r$ as $J \sim 1/r^{\nu}$ with the resonant dipole-dipole interaction, $\nu = 3$, being most frequently the case [1,2,5–7,11].

Many of such systems are inherently disordered. Diagonal disorder leads to exponential (Anderson) localization of all the eigenstates of one-dimensional systems [12–14], which would suppresses excitation transfer in sufficiently long spin chains. Off-diagonal disorder also leads to localization which, however, may be weaker than exponential [15–17]. The localization properties of the system with long-range exchange interaction are more subtle [18–23] and many features still merit further investigation, which is one of the motivations of the present work.

Specifically, we study disordered spin chains – collection of two-level atoms, molecules or spins arranged in nearly periodic quasi one-dimensional array and coupled with each other by the resonant dipole-dipole exchange interaction. We raise the questions whether or not, and to what degree, such a disordered system can serve for excitation or spin polarization transfer between the sender and the receiver spins coupled to the opposite ends of the chain in a controllable way. To that end, we first determine the (single-excitation) localization properties of the system and their dependence on the energy, comparing and contrasting spin systems with long-range and nearest-neighbor exchange interactions. Obviously, only chains of length smaller or comparable to the longest localization length can transfer excitation between the two ends. Next we identify the energy eigenstates that have sufficient support at the two ends of the chain to strongly couple to the sender and receiver spins. We then explore two excitation transfer protocols, one that involves static resonant couplings of the sender and receiver spins to the most suitable eigenstate of the chain [24,25,25], and the other inspired by stimulated Raman adiabatic transfer [26–28] that involves counterintuitive time-dependent couplings of the sender and receiver spins to the corresponding eigenstate of the chain. We find that the adiabatic coupling, despite being slower than the static coupling scheme, leads to a much higher probability of excitation transfer as it is more robust to various sources of disorder.

The paper is organized as follows. In Sec. 2 we introduce the Hamiltonian of the system involving a collections of spins (two-level systems) with long-range resonant dipole-dipole exchange interactions and formulate the transfer problem. In Sec. 3 we consider disordered spin chains and numerically determine the localization lengths for different single-excitation eigenstates of the system in the presence of energy (diagonal) and position (off-diagonal) disorder. In Sec. 4 we present two excitation transfer protocols between the sender and receiver spins resonantly coupled to a suitable energy eigenstate of a spin chain with no disorder. In Sec. 5 we extract the mean transfer probability for chains of different length with different strength and type of disorder. Our conclusions are summarized in Sec. 6.

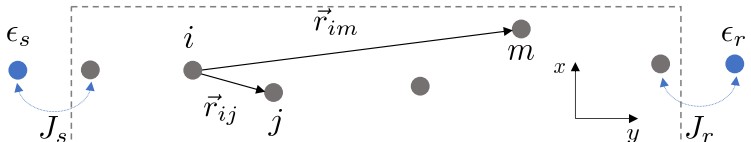

Figure 1: Schematic of a position (and energy) disordered chain of spins $i, j, \ldots, m, \ldots$ in the $xy$ plane. The spin chain is coupled with rates $J_{s,r}$ to the sender (s) and receiver (r) spins having energies $\epsilon_{s,r}$.

## 2 The system

We consider a chain of $N$ spins – two-level systems – interacting with each other via the long-range exchange interactions $J_{ij} = C_3(1 - 3\cos^2\theta_{ij})/\left|\vec{r}_{ij}\right|^3$, where $C_3 \propto |\vec{\wp}|^2$ is the electric or magnetic dipole-dipole interaction coefficient, $\vec{r}_{ij}$ is the position vector between spins $i$ and $j$, and $\theta_{ij}$ is the angle between the direction of the dipole moments $\vec{\wp}$ and the position vector between the spins. We account only for the near-field part of the total dipole-dipole interaction potential and neglect the retardation and spontaneous radiative decay of the spin excitations [29–31], assuming that the typical distance between the spins is much smaller than the wavelength of the transition between the spin-up and spin-down states. The Hamiltonian of the system is

$$\mathcal{H} = \frac{1}{2}\sum_{i=1}^{N}\epsilon_i\hat{\sigma}_i^z + \sum_{i\neq j}^{N}J_{ij}(\hat{\sigma}_i^+\hat{\sigma}_j^- + \hat{\sigma}_j^+\hat{\sigma}_i^-),\tag{1}$$

where $\epsilon_i$ is the excitation energy of spin $i$, $\hat{\sigma}_i^{x,y,z}$ are the Pauli spin operators and $\hat{\sigma}_i^{\pm} = \frac{1}{2}(\hat{\sigma}_i^x \pm i\hat{\sigma}_i^y)$ are the raising and lowering operators. We assume that all the spins are positioned in one ($xy$) plane (see Fig. 1) and their dipole moments ($\vec{\wp} \parallel \hat{z}$) are perpendicular to that plane, $\theta_{ij} = \pi/2 \; \forall \; i,j$, thus $J_{ij} = C_3/\left|\vec{r}_{ij}\right|^3$.

We assume that a sender and a receiver spins are coupled in controllable way to the opposite ends of a finite spin chain, see Fig. 1. The spin chain is assumed initially fully polarized, with all the spins unexcited. Our aim is to transfer an excitation between the sender and receiver spins via the spin chain. To this end, we need to identify and employ extended eigenstates of the disordered chain having sufficient support at its two ends in order to strongly couple to sender and reciever spins and mediate the transfer. To selectively couple the sender and receiver spins to the suitable eigenstates of the chain, we assume that their energies $\epsilon_s$, $\epsilon_r$ and couplings $J_s$, $J_r$ to the first and last spins of the chain can be precisely controlled, unlike the energies and couplings of the spins in the disordered chain. Initially, the excitation is localized at the sender spin, while the spin chain contains no excitations, and our aim will be to retrieve the excitation from the receiver spin at a specific time $\tau$ to be determined below.

We next examine the localization length of the single-excitation eigenstates of spin chains in the presence of diagonal disorder corresponding to energy disorder of individual spins, and off-diagonal disorder in the interspin couplings stemming from the position disorder of the spins.

## 3 Localization lengths in disordered spin chains

We impose diagonal disorder corresponding to random variations of the spin excitation energies $\epsilon_j = \epsilon_0 + \delta\epsilon_j$ around some $\epsilon_0$ (which can be set to 0) with $\delta\epsilon_j$ having a Gaussian probability distribution $P(\delta\epsilon) = \frac{1}{\sqrt{2\pi\sigma_\epsilon^2}}e^{-\frac{\delta\epsilon^2}{2\sigma_\epsilon^2}}$ with the mean $\langle\delta\epsilon\rangle = 0$ and variance $\sigma_\epsilon^2$. Next,

the position of each spin $j$ is given by the coordinates $(x_j, y_j)$. In an ideal 1D lattice with period $a$, we would have $x_j = aj$ and $y_j = 0$ for all spins $j = 1, 2, \ldots, N$, and the exchange interaction strength between the nearest-neighbor spins would be $J = C_3/a^3$, the next-nearest neighbors $J/2^3$, etc. We impose the position disorder via $x_j \to aj + \delta x_j$ and $y_j \to \delta y_j$, where the random variables $\delta x_j$ and $\delta y_j$ have a Gaussian probability distribution $P(\delta\mu) = \frac{1}{\sqrt{2\pi\sigma_\mu^2}} exp(-\frac{\delta\mu^2}{2\sigma_\mu^2})$ ($\mu = x, y$) around mean $\langle\delta\mu\rangle = 0$ with variance $\sigma_\mu^2$. The position disorder then translates to off-diagonal (interspin coupling) disorder in the Hamiltonian (1).

In the limit of $N \to \infty$, disorder leads to (Anderson) localization of all the eigenstates of the system [12–14]. The wavefunction $\psi_k(x)$ of each single-excitation eigenstate $|\psi_k\rangle$ is then localized around some position $\mu_k$ with the localization length $\xi_k$. An important characteristic of the system is the dependence of the localization length $\xi_k$ on the energy $E_k$ of the eigenstates to be used for the excitation transfer. To determine the localization length, we numerically diagonalize the Hamiltonian for sufficiently long chains ($N = 1000$ spins) to neglect the finite size effects, and then for each eigenstate we identify the position $\mu_k$ corresponding to the maximum (in absolute value) of the wavefunction $\psi_k(x)$ and subsequently fit an exponential function

$$|\psi_k(x)| \propto e^{-\frac{|x-\mu_k|}{\xi_k}}, \tag{2}$$

to the spatial profile of the eigenstate, extracting thereby the localization length $\xi_k$. We note that the thus obtained localization length is a convenient measure of the spatial extent of the wavefunction even if it is not exponentially localized (see below).

A more common measure to quantify the localization properties of the eigenstates is the inverse participation ratio (IPR) [32]. It is, however, not suitable for our purposes, since IPR cannot determine whether a wavefunction is spatially localized on a number of neighboring sites or is delocalized on a similar number of remote sites.[1] We use, therefore, an alternative method to verify that the localization length $\xi_k$ extracted from the exponential fit (2) is a reliable quantity to characterize our system. We can partition the chain into two halves and for each eigenstate $|\psi_k\rangle = \sum_{i=1}^{N} v_i^{(k)} |i\rangle$ calculate the excitation number variance in one of the halves [35],

$$\Delta n_k^2 = \langle \hat{n}^2 \rangle - \langle \hat{n} \rangle^2, \tag{3}$$

where $\hat{n} = \sum_{i=1}^{N/2} \hat{\sigma}_i^+ \hat{\sigma}_i^-$ is the excitation number operator with eigenvalues $n = 0, 1$ since we consider only single-excitation states. The variance is therefore given by

$$\Delta n_k^2 = p_k - p_k^2, \tag{4}$$

where $p_k = \sum_{i=1}^{N/2} |v_i^{(k)}|^2$ is the probability to find the excitation in the left half of the chain.

Clearly, for a strongly localized state with $\xi/a \ll N/2$, the probability $p$ is either close to 0 or close to 1 (unless the wavefunction is localized near the center of the chain, $\mu/a \simeq N/2$, the probability of which is $2\xi/(aN) \ll 1$), and the number variance is small, $\Delta n^2 \to 0$. In the opposite limit of a completely delocalized wavefunction $\xi/a > N$, the probability is $p \simeq 1/2$ and the number variance approaches the maximum $\Delta n^2 \to 1/4$. Assuming an exponentially localized wavefunction $\psi(x)$ of the form (2), we can calculate $p$ for any position of the peak $\mu$, and upon averaging over the peak positions $\mu/a \in [1, N]$ we obtain a relation

---

[1]The inverse participation ratio [32] IPR $= \sum_i |v_i|^4$ for a wavefunction $|\psi\rangle = \sum_{i=1}^{N} v_i |i\rangle$ quantifies on how many lattice sites $i$ the wavefunction has support, i.e., IPR is small for a uniformly delocalized wavefunction, $|v_i|^2 \sim 1/N \, \forall i$, and is large if many sites have vanishing populations. The latter, however, does not mean that the wavefunction is spatially localized, because a wavefunction having large populations on only a few lattice sites separated by large distance from each other would also have a large IPR. This is in fact what we observe for lattices with long-range exchange interactions and off-diagonal disorder. Similar wavefunction bi-localization phenomena also occur in other disordered lattice systems [33, 34].

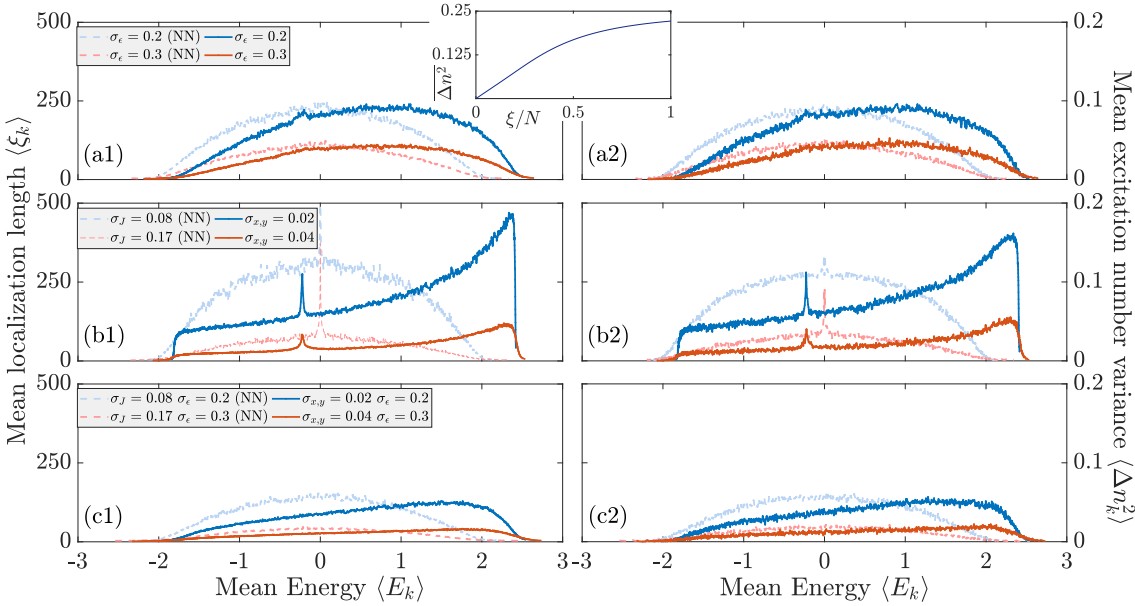

Figure 2: Mean localization length $\langle \xi_k \rangle$ (in units of lattice spacing $a = 1$) [left panels (a1), (b1), (c1)], and mean excitation number variance $\langle \Delta n_k^2 \rangle$ [right panels (a2), (b2), (c2)] vs the mean energy $\langle E_k \rangle$ (in units of $J = C_3/a^3$) of the $k$-th eigenstate of a chain of $N = 1000$ spins obtained upon averaging over 1000 independent realizations of disordered chains with long-range exchange interactions (solid lines with filled circles) and nearest-neighbor interactions (dashed lines), for (a) energy (diagonal) disorder with standard deviation $\sigma_\epsilon$, (b) position (off-diagonal) disorder with standard deviation $\sigma_{x,y}$ or $\sigma_J$, and (c) combination of energy and position disorder. For illustrative purposes, we use in (a) and (b) the strength of the diagonal $\sigma_\epsilon$ and off-diagonal $\sigma_{x,y}$ (or $\sigma_J$) disorders that lead to comparable localization lengths. Inset shows the averaged number variance $\overline{\Delta n^2}$ vs $\xi/N$, as described in the text.

between $\overline{\Delta n^2}$ and $\xi/N$ shown in the inset of Fig. 2. For small $\xi/a < N/2$, the number variance grows approximately linearly with the localization length as $\overline{\Delta n^2} \approx \frac{3}{8} \frac{\xi}{aN}$, and it starts to saturate thereafter. We note that an equivalent measure of localization of a single-excitation wavefunction in one partition of the system is the entanglement entropy $S$ [35, 36] related to the number variance via $S \geq (4 \ln 2) \Delta n^2$.

In Fig. 2 (left panels: a1, b1, c1), we show the mean localization length $\langle \xi_k \rangle$ versus the mean energy $\langle E_k \rangle$ of the eigenstates of spin chains with long-range exchange interactions for three different cases: (a) diagonal (energy) disorder, (b) off-diagonal (position) disorder, and (c) combination of diagonal and off-diagonal disorders. The corresponding mean excitation number variances $\langle \Delta n_k^2 \rangle$ are shown in Fig. 2 (right panels: a2, b2, c2). For each case we consider two different strengths of the disorder determined by the standard deviations $\sigma_\epsilon$ and $\sigma_{x,y}$.

For comparison, we also consider chains with nearest-neighbor exchange interactions and the same effective disorder as described by Hamiltonian

$$\mathcal{H}_{\text{nn}} = \frac{1}{2} \sum_{i=1}^{N} \epsilon_i \hat{\sigma}_i^z + \sum_{i=1}^{N-1} J_i (\hat{\sigma}_i^+ \hat{\sigma}_{i+1}^- + \hat{\sigma}_{i+1}^+ \hat{\sigma}_i^-), \tag{5}$$

where $\epsilon_i$ are the random spin energies as above, while $J_i = J + \delta J_i$ are the exchange couplings with $J = C_3/a^3$ and $\delta J_i$ being Gaussian random variables with the mean $\langle \delta J \rangle = 0$ and standard

deviation determined by the uncertainty propagation formula

$$\sigma_J \approx |\partial_x D(x,y)|\sigma_x + |\partial_y D(x,y)|\sigma_y,$$

where $D(x,y) = C_3/(x^2+y^2)^{3/2}$.

Note that, in an ideal lattice with no disorder, the single excitation spectrum of Hamiltonian (1) is given by

$$E_k = 2\sum_{m=1}^{N} \frac{J}{m^3} \cos\frac{\pi k m}{N+1}, \tag{6}$$

while the spectrum of the system with only the nearest-neighbor interactions, Eq. (5), corresponds to the $m=1$ term in the above sum, i.e. $E_k^{(\mathrm{nn})} = 2J\cos\frac{\pi k}{N+1} \in [-2J, 2J]$. One can treat perturbatively the $m>1$ terms of Eq. (6) near the band edges and deduce [37, 38] that the lower edge of the energy band is shifted from $-2J$ to approximately $-1.8J$ while upper edge is shifted from $2J$ to approximately $2.4J$. Thus, the long-range character of the exchange interaction affects the energy band structure and the density of states.

**Diagonal disorder.** Consistent with the above discussion, for a chain with long-range exchange interactions and diagonal disorder, we observe in Fig. 2(a1) and (a2) that the profile of the mean localization length $\langle \xi_k \rangle$ and the nearly identical profile of the mean excitation number variance $\langle \Delta n_k^2 \rangle$ are shifted and skewed towards the higher energies $\langle E_k \rangle$, as compared to the nearest-neighbor interacting chains. For the presently considered dipole-dipole interactions, $J_{ij} \propto 1/|r_{ij}|^3$, the localization length $\langle \xi_k \rangle$ remains finite for all energies $\langle E_k \rangle$. We note, however, that for power-law interaction $J_{ij} \propto 1/|r_{ij}|^\nu$ with decreasing $\nu$ a localization-delocalization transition occurs at $\nu = 3/2$ near the (shifted) upper edge of the energy band $\langle E_k \rangle \approx 5J$ [39].

**Off-diagonal disorder.** Even though the wavefunctions of the eigenstates of a chain with off-diagonal disorder may not be exponentially localized for all energies, for consistency and comparison with diagonal disorder, we still use the exponential fit of Eq. (2) to deduce the localization length and verify its applicability by the corresponding excitation number variance. For the nearest-neighbor interacting chain with only off-diagonal disorder, the first feature to note in Fig. 2(b1, b2) is the sharp peak of the localization length at zero energy. This peak is related to the well-known divergence of the density of states $\rho(E) \sim \frac{1}{E|\ln E|^3}$ [40, 41] leading to the localization length divergence as $\xi \sim |\ln E|$ that follows from the Thouless relation [42]. But unlike the case of diagonal disorder, the eigenstates near zero energy are localized as $|\psi(x)| \propto e^{-\sqrt{x/\zeta}}$ rather than exponentially [15–17]. We note the relevant early studies of Dyson [43][2] and the insightful connection to the graph theoretical concepts [16].[3]

The long-range exchange interactions in the chain with off-diagonal disorder [38, 44, 45] lead to certain modification of the localization spectrum. The zero-energy peak of the nearest-neighbor interacting chain is now displaced to $\langle E_k \rangle \simeq -0.22J$, which follows from the perturbative treatment of Eq. (6) near the center of the band [38], and is suppressed, since

---

[2]The study of the anomalous behavior of the localization length near zero energy was initiated by the work of Dyson [43] on the one-dimensional random harmonic oscillator chain with Poisson distributed couplings. The singularity exhibits universal behavior as long as the probability distribution of the couplings is well behaved.

[3]The sharp peak of the localization length at zero energy in lattices with off-diagonal disorder has been connected to the bipartite nature of the lattice [16]. A lattice is called bipartite if the vertices can be partitioned in two independent and disjoint sets such that every edge connects vertices that belong to a different set. This peak is suppressed when the underlying lattice is not bipartite, i.e., in the presence of next-nearest neighbor interactions or diagonal disorder. In our case, the next-nearest neighbor interactions $J/8$ make the underlying lattice only weakly non-bipartite, and the peak is shifted and suppressed, but still survives.

the underlying lattice is weakly non-bipartite due to the weak next-nearest-neighbor interactions, which is in complete agreement with our numerical results in Fig. 2(b1, b2). We note again that the use of IPR is inadequate to quantify the localization length in the vicinity of $\langle E_k \rangle \simeq -0.22J$, as it would indicate more, rather than less, localized states [38]. That is why we still use the localization length $\langle \xi_k \rangle$ obtained from the exponential fit of Eq. (2) and verify its applicability by the corresponding excitation number variance $\langle \Delta n^2 \rangle$.

Another feature is that, perhaps counterintuitively, disordered chains with long-range exchange interactions exhibit shorter localization length in the central part of the spectrum, as compared to chains with only nearest-neighbor interactions [18, 20, 21]; in effect the long-range interactions amplify the disorder. But for larger energies the localization length $\langle \xi_k \rangle$ (and the excitation number variance $\langle \Delta n_k^2 \rangle$) gradually increases [38, 46] and it exhibits a sharp peak near the upper edge of the energy band, $\langle E_k \rangle \approx 2.4J$. The states near the upper edge of the energy band are in fact completely delocalized, $\langle \xi_k \rangle \approx N/2$, at least for not too strong off-diagonal disorders that we consider. This behaviour is reminiscent to the emergence of extended states at the band edge for spin chains with diagonal disorder and long-range hopping $J_{ij} \propto 1/|r_{ij}|^\nu$ with decreasing power $\nu$, but for our case of off-diagonal disorder and $\nu = 3$, the sharp peak is much more pronounced.

**Combined diagonal and off-diagonal disorder.** Finally in Fig. 2(c1, c2) we show the mean localization length and the mean excitation number variance versus the mean energy for the chains with both diagonal and off-diagonal disorders that concurrently localize the system eigenstates. Now the (shifted) zero-energy peak is completely suppressed[4] while the eigenstates with the longest localization length reside between the center and the upper edge of the band skewed by the long-range exchange interactions.

To summarize, the important information gained by our analysis of the localization lengths in disordered spin chains is the maximum length of a finite chain that can support excitation transfer through an extended eigenstate. Conversely, when the chain length exceeds the maximum localization length of the eigenstates, we expect the transfer to be completely suppressed. We note that in all cases when the obtained mean localization length is sufficiently shorter than the chain length, $\langle \xi_k \rangle < aN/2$, the relation $\langle \Delta n_k^2 \rangle \approx \frac{3}{8} \frac{\langle \xi_k \rangle}{aN}$ holds to a very good approximation, which justifies our approach to characterizing the localization properties of disordered spin chains with long-range exchange interactions.

## 4 Excitation transfer schemes

The large localization length of single-excitation eigenstates in a spin chain is necessary but not yet sufficient to ensure efficient transfer of excitation between the sender and receiver spins. Rather, the extended eigenstates of the chain should have sufficient support at the two ends of the chain in order to strongly couple to the sender and receiver spins.

To illustrate the procedure for excitation transfer, in this section we consider spin chains with long-range interactions but no disorder. Solving the eigenvalue problem

$$\mathcal{H} |\psi_k\rangle = E_k |\psi_k\rangle , \tag{7}$$

we obtain the eigenstates $|\psi_k\rangle = \sum_i v_i^{(k)} |i\rangle$ which couple to the sender and receiver spins at the two ends of the chain with the corresponding strengths

$$\Omega_s^{(k)} = J_s v_1^{(k)} , \quad \Omega_r^{(k)} = J_r v_N^{(k)} , \tag{8}$$

---

[4]The diagonal disorder suppresses the shifted zero-energy (Dyson) peak since the varying onsite energies are equivalent to self-interaction of the vertices which violates the bipartition of the lattice [16].

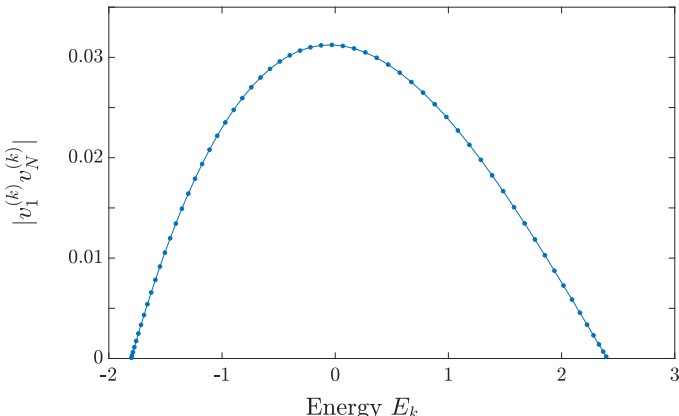

Figure 3: Absolute value of the product $|v_1^{(k)} v_N^{(k)}|$ of the boundary amplitudes of $k$-th eigenvector of the chain vs the eigenenergy $E_k$ (in units of $J$), for an ordered chain of $N = 61$ spins with long-range exchange interaction.

where $J_s$ and $J_r$ are the coupling strength of the sender and receiver spins to the first and the last spins of the chain. Hence, in order to efficiently transfer the excitation from the sender to the receiver spin via a particular eigenstate $|\psi_k\rangle$ of the chain, this eigenstate should have large amplitudes $|v_{1,N}^{(k)}|$ at both ends of the chain.

In Fig. 3 we show the absolute value of the product $|v_1^{(k)} v_N^{(k)}|$ of the boundary amplitudes of the different energy eigenstates $|\psi_k\rangle$ of the chain. This figure reveals that the eigenstates most suitable for the transfer are in the middle of the spectrum, $E_k \sim 0$, while the eigenstates at the upper edge of the spectrum, $E_k \lesssim 2.4J$, would only weakly couple to the sender and receiver spins and are thus unsuitable for the excitation transfer, despite having large (or even divergent) localization length in disordered chains. Having in mind the chains with both diagonal and off-diagonal disorder exhibiting the localization peak in the vicinity of $E = -0.22J$, we shall tune the energies of the sender and receiver spins to $\epsilon_{s,r} \approx -0.22J$.

Another critical issue for the efficient transfer via the selected eigestates of the chain is the small leakage of the excitation, initially at the sender spin, to all other non-resonant eigenstates of the chain [24, 25]. In a chain of $N$ spins, the average distance between the energy eigenstates is $\Delta E \simeq 4J/N$. Therefore the coupling strength of the sender and receiver spins, tuned to resonance to a particular eigenstate, should satisfy $\Omega_{s,r} < \Delta E$. Since the amplitudes of the edge states for the most delocalized eigenstates are $v_{1,N}^{(k)} \sim 1/\sqrt{N}$, we obtain from (8) that the coupling rates should satisfy $J_{s,r} \lesssim J/\sqrt{N}$ in order to avoid the leakage of the excitation to the undesired states of the chain and attain high transfer probability [47].

**Static coupling to the chain.** To illustrate the ongoing discussion, in Fig. 4 we show the dynamics of excitation transfer between the sender and receiver spins via spin chains of different length $N$ with no disorder. For convenience, we chose chains with odd number of spins, $N = 11, 21, \ldots$, and tune the energies of the sender and receiver spins $\epsilon_{s,r}$ to the energy of the "fittest" eigenstate closest to $E = -0.22J$.

The state of the system in the single excitation subspace can be written as $|\Psi\rangle = \alpha_s |s\rangle + \sum_{i=1}^{N} \alpha_i |i\rangle + \alpha_r |r\rangle$, where $\alpha_j$ are the amplitudes and $|j\rangle$ denotes the state with the excitation at position $j = s, r$ or $i \in [1, N]$. Initially the excitation is localized at the sender spin, $|\Psi(0)\rangle = |s\rangle$, and the couplings $J_{s,r}$ are set to the constant values $J_{s,r} \simeq 0.5J/\sqrt{N}$. The state of the system $|\Psi(t)\rangle$ evolves according to the Hamiltonian (1), and the transfer probability to the receiver spin $P_r(t) = |\langle r|\Psi(t)\rangle|^2$ is shown in Fig. 4(a). In a three-state system,

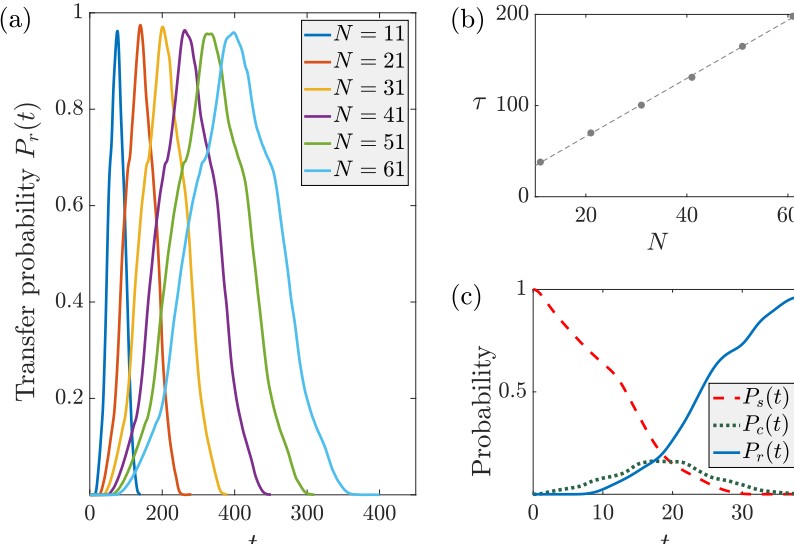

Figure 4: Excitation transfer via static couplings of the sender and receiver spins with rates $J_{s,r} = 0.49 J/\sqrt{N}$ to the chain of $N$ spins with long-range exchange interactions and no disorder. (a) Transfer probability $P_r(t)$ vs time $t$ (in units of $1/J$) for different chain lengths $N$. The energies of the sender and receiver spins $\epsilon_{s,r}$ are tuned to the energy of the eigenstate of the chain closest to $E = -0.22J$. (b) Transfer time $\tau$ (gray filled circles), corresponding to the first peak of the transfer probability in (a) for each chain length $N$. Dashed line shows the linear fit $\tau J = 3.2N + 2.3$. (c) Time-evolution of the excitation probability for the sender $P_s(t)$, receiver $P_r(t)$ and intermediate chain $P_c(t)$, for a chain of $N = 11$ spins.

complete transfer would occur at time $\tau = \pi/(2\sqrt{2}\Omega_{s,r})$. Our multilevel system now behaves as an effective three-state system with a single intermediate eigenstate of the chain, and the transfer time scales as $\tau \propto N$ consistently with $\Omega_{s,r} \propto 1/N$, see Fig. 4(b). We note that the linear scaling of the transfer time with the length of the chain stems from our requirement to avoid the leakage of excitation into the undesired states. But this scaling is consistent with the Lieb-Robinson bound [48–50] which is in fact a much lower bound for the achievable transfer time although with a reduced transfer probability due to the leakage. In Fig. 4(c) we show the dynamics of probabilities of excitation of the sender spin, $P_s(t) = |\langle s|\Psi(t)\rangle|^2$, the chain, $P_c(t) = \sum_{i=1}^{N} |\langle i|\Psi(t)\rangle|^2$, and the receiver spin, $P_r(t)$, during one full transfer cycle.

**Time-dependent adiabatic couplings.** In a three-state system, a more efficient excitation transfer can be achieved using an analog of stimulated Raman adiabatic passage (STIRAP) [26–28]. It involves time-dependent couplings and must be sufficiently slow in order to be adiabatic, but is robust and avoids populating the intermediate – here the spin-chain – state(s).

Consider an effective three-state system $|\Psi\rangle = \alpha_s |s\rangle + \alpha_k |\psi_k\rangle + \alpha_r |r\rangle$ governed by the Hamiltonian

$$\mathcal{H}^{\text{eff}} = \Delta\epsilon_k |\psi_k\rangle \langle\psi_k| + (\Omega_s^{(k)} |s\rangle \langle\psi_k| + \Omega_r^{(k)} |r\rangle \langle\psi_k| + \text{H.c.}), \qquad (9)$$

where $\Delta\epsilon_k = E_k - \epsilon_{s,r}$ is a possible energy mismatch between the selected eigenstate of the chain $|\psi_k\rangle$ and the sender and receiver spins. This Hamiltonian has a zero-energy coherent population trapping (or dark) eigenstate $|\Psi_0\rangle \propto \Omega_r^{(k)} |s\rangle - \Omega_s^{(k)} |r\rangle$ that does not involve the intermediate state $|\psi_k\rangle$ of the spin chain. With the excitation initially localized on the sender spin, we set the coupling $|\Omega_r^{(k)}| \gg |\Omega_s^{(k)}|$ such that the dark state coincides with the initial state, $|\Psi_0\rangle = |s\rangle$. We then slowly switch off $\Omega_r^{(k)}$ and switch on $\Omega_s^{(k)}$, which results in an adiabatic

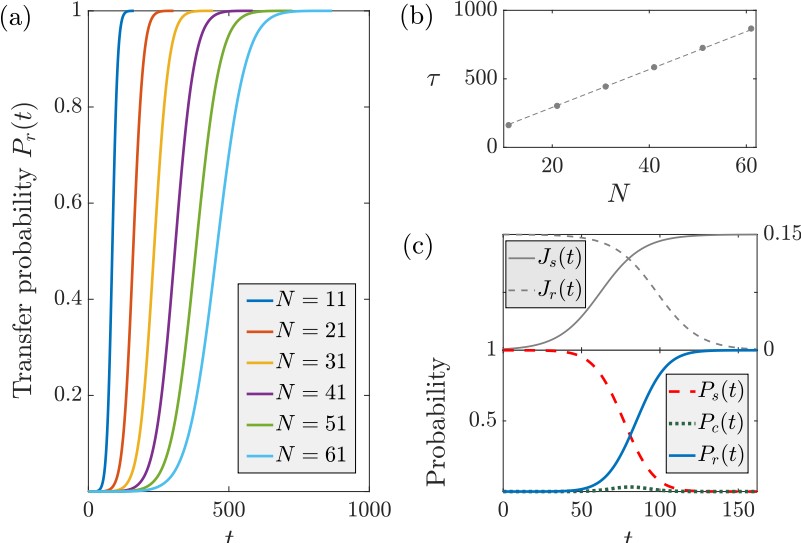

Figure 5: Stimulated adiabatic transfer of excitation between the sender and receiver spin using time-dependent couplings of Eq. (10), for chains with long-range exchange interactions and no disorder. (a) Transfer probability $P_r(t)$ vs time $t$ (in units of $1/J$) for chains of different length. (b) Transfer time $\tau$ (gray filled circles) as a function of $N$, and the linear fit $\tau J = 14.1N + 6.9$ (dashed line). (c) Top panel shows the time-dependent coupling rates $J_{s,r}(t)$ of Eq. (10), and the bottom panel shows the dynamics of excitation probabilities of the sender $P_s(t)$, receiver $P_r(t)$ and intermediate chain $P_c(t)$, for $N = 11$.

rotation of the dark state $|\Psi_0\rangle$ towards $|r\rangle$, and at the final time $\tau$, when $|\Omega_r^{(k)}| \ll |\Omega_s^{(k)}|$, we obtain $|\Psi_0\rangle \simeq |r\rangle$. To realize this so-called counterintuitive pulse sequence, we use the time-dependent boundary couplings

$$J_{s,r}(t) = \frac{J_{s,r}^{\max}}{2}\Big(1 \pm \tanh\left(\gamma t/\tau - \beta_{s,r}\right)\Big), \tag{10}$$

where $J_{s,r}^{\max} \simeq 0.5/\sqrt{N}$ as before, while the parameters $\gamma = 6$, $\beta_{s,r} = 2.3, 3.6$ and the process duration $\tau \propto N$ are chosen so as to optimize the overlap between the pulses and achieve adiabaticity with sufficiently large effective pulse area $\int_0^\tau dt \sqrt{|\Omega_s^{(k)}(t)|^2 + |\Omega_r^{(k)}(t)|^2} \gtrsim 10$ [27, 28]. We note that the adiabatic population transfer has been applied to multilevel systems before [51, 52].

In Fig. 5 we illustrate the adiabatic transfer protocol for ordered chains of different length and time-dependent couplings of Eq. (10) but otherwise the same parameters as in Fig. 4. We achieve nearly perfect population transfer for all considered cases, see Fig. 5(a), at the expense of longer duration of the process $\tau$, see Fig. 5(b). Note that during the transfer, as the system adiabatically follows the coherent population trapping state $|\Psi_0\rangle$, the chain contains almost no excitation at all times, Fig. 5(c).

## 5 Transfer probability in disordered chains

Having determined the localization lengths $\xi$ in long disordered spin chains in Sec. 3 and potentially suitable excitation transfer protocols in Sec. 4, we now analyze the mean probability $\langle P_r \rangle$ of excitation transfer between the sender and receiver spins via disordered spin chains of finite length $N$ comparable to $\xi$.

**Static coupling to the chain.** We first consider the static transfer protocol of Fig. 4 with fixed coupling rates $J_{s,r} \simeq 0.5J/\sqrt{N}$ of the sender and receiver spins having energies $\epsilon_{s,r} = -0.22J$. With the excitation initially localized at the sender spin, we terminate the evolution when the excitation probability of the receiver spin attains its first maximum at $t = \tau$ of Fig. 4(b). In Fig. 6 we show the transfer probabilities $\langle P_r \rangle$ averaged over many independent realizations of disordered spin chains, involving spin-energy (diagonal) disorder, spin-position (off-diagonal) disorder, and the combination of the two. As expected, increasing the chain length $N$ decreases the transfer probability which is due to the stronger disorder-induced localization of the eigenstates of the chain in the middle of the energy spectrum. We also observe that chains with only the nearest-neighbor exchange interaction (with $\epsilon_{s,r} = 0$) lead to better transfer probability, especially for the case of off-diagonal disorder, Fig. 6(b), which is consistent with their larger localization length under otherwise similar conditions, as discussed in Sec. 3 and seen in Fig. 2(b).

**Time-dependent adiabatic couplings.** We finally consider the adiabatic transfer protocol of Fig. 5 with the time-dependent coupling rates of Eq. (10) applied to the sender and receiver spins in a counterintuitive order. In Fig. 7 we show the results of our numerical simulations for the transfer probabilities $\langle P_r \rangle$ averaged over many independent realizations of disordered spin chains. Compared to the static transfer protocol, the performance of adiabatic transfer is significantly better for all chain lengths and any kind of disorder, be it diagonal, off-diagonal, or combination of both. We emphasize that in this study, we have focused on the spin excitation or polarization transfer probability. In contrast, coherent quantum state transfer is much more

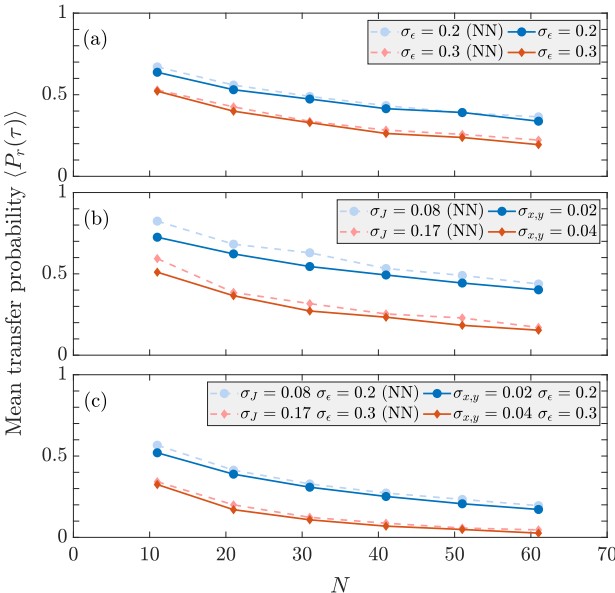

Figure 6: Mean excitation transfer probability $\langle P_r \rangle$ vs chain length $N$, obtained upon averaging over 1000 independent realizations of disordered chains with long-range exchange interactions (solid lines with filled circles and diamonds) and nearest-neighbor interactions (dashed lines with light filled symbols), for (a) energy (diagonal) disorder with standard deviations $\sigma_\epsilon$, (b) position (off-diagonal) disorder with standard deviation $\sigma_{x,y}$ or $\sigma_J$, and (c) combination of energy and position disorder. We use the static couplings of the sender and receiver spins $J_{s,r} = 0.49J/\sqrt{N}$ having energies $\epsilon_{s,r} = -0.22J$ ($\epsilon_{s,r} = 0$ for the nearest-neighbor interacting chains), and the evolution is terminated at $t = \tau$ of Fig. 4(b).

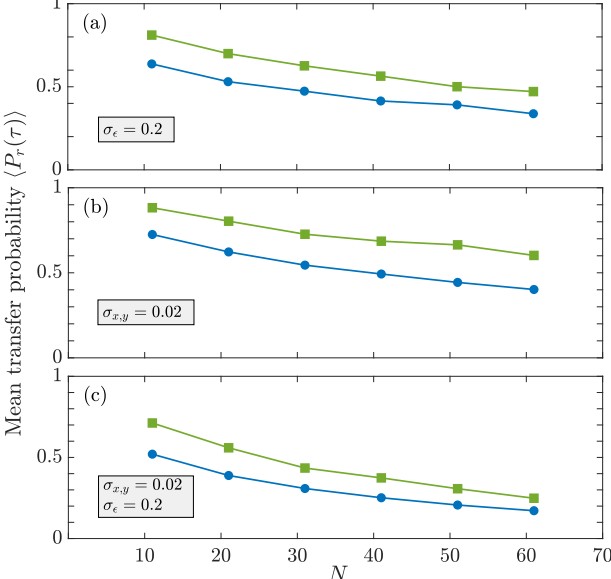

Figure 7: Mean excitation transfer probability $\langle P_r \rangle$ vs chain length $N$, obtained upon averaging over 1000 independent realizations of disordered chains with long-range exchange interactions, for the stimulated adiabatic transfer (green solid lines with filled squares), compared to the static transfer of Fig. 6 (blue solid lines with filled circles), for (a) energy (diagonal) disorder with standard deviations $\sigma_\epsilon$, (b) position (off-diagonal) disorder with standard deviation $\sigma_{x,y}$, and (c) combination of energy and position disorder. We use the time-dependent couplings of Eq. (10) for the sender and receiver spins having energies $\epsilon_{s,r} = -0.22J$, with the transfer duration $\tau$ of Fig. 5(b).

sensitive to diagonal disorder leading to larger dephasing during adiabatic transfer which is necessarily slower than the static transfer [53].

## 6 Conclusions

We have presented the results of our studies of disordered, one-dimensional spin-chains with long-range exchange (resonant dipole-dipole) interactions and their ability to transfer spin excitation or polarization over long distances. We have performed detailed numerical investigations of the localization length in spin chains with either or both diagonal and off-diagonal disorder. Many of our results concur with the previously known and well-understood properties of disordered spin chains, but we have also encountered interesting manifestations of (de)localization of energy eigenstates that, to the best of our knowledge, have not been properly addressed before in the context of resonant dipole-dipole ($1/r^3$) interactions, and thus may warrant further investigation. These, in particular, include delocalization of the eigenstates at the upper edge of the shifted energy band in spin chains with off-diagonal disorder, and the modification of the shifted zero-energy Dyson peak of localization length, which we found to be the most suitable eigenstate for the excitation transfer between the two ends of the chain.

We have put forward two excitation transfer protocols: a) static protocol involving selective coupling of the sender and receiver spins to the suitable eigenstate of the chain, and b) time-dependent adiabatic protocol involving counter-intuitive sequence of couplings of the sender

and receiver spins to the chain, inspired by stimulated Raman adiabatic passage technique widely used in atomic and molecular physics. We have found that the adiabatic transfer of excitation via disordered spin chains has much better performance for all chain length and any kind of disorder, be it diagonal, off-diagonal, or combination of both. This attests, once again, the usefulness of this universal method.

## Acknowledgements

We thank Ivan Khaymovich for helpful comments and suggestions.

**Funding information**

This work was supported by the EU QuantERA Project PACE-IN (GSRT Grant No. T11EPA4-00015 for N. E. P. and D. P.). M.K-E. acknowledges financial support from Deutsche Forschungsgemeinschaft (DFG) via SFB TR 185, Project No.277625399. G. K. acknowledges also the support of PATHOS (FET Open) and DFG (FOR 7274).

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
