# Peer review of "Excitation transfer in disordered spin chains with long-range exchange interactions"

_SciPost Physics Core, doi:SciPost Phys. Core 6, 017 (2023)_

## Round 1 · Referee Report · Anonymous (Referee 1) · 2022-10-13

Strengths

1.- The topic is very interesting and timely.

2.- The analysis of excitation transfer in disordered chains seems to pose very interesting questions.

Weaknesses

1.- First of all, the authors claim that the XX model is "interacting" in the abstract itself. Yet, all their discussion is done in terms of single-particle modes. These claims are not compatible. Of course, I guess that they are assuming that the XX model has been re-written as a free fermionic Hamiltonian using the Jordan-Wigner transformation. Moreover, I have several crucial issues in this respect.

First of all, the authors should make their claims explicit. If they start out with an XX model and convert it to a free-fermionic model, they should say so.

2.- The long-range XX model can not be converted into a long-range quadratic free-fermionic Hamiltonian through the Jordan-Wigner transformation. The reason is the non-locality of the fermionic operators, which only cancels out in the 1D case.

3.- Even if we were using the short-ranged XX model, I still do not have a clear idea of how they build the quantum states. Indeed, the ground state of the XX model can be formed in this case as a Fermi sea (or Dirac vacuum, if you prefer). Perhaps they are focusing on the single-particle sector, i.e. nearly-maximal magnetization in the XX-model. If so, they should explain it, but I expect this is not the case, because the results would be far less interesting.

4.- The authors base their analysis of the wavefunctions assuming that they all fit the exponential ansatz, eq. (2). Yet, this is known not to be the case for some of the studied cases. Indeed, for local off-diagonal disorder, the single-body modes are known to be bi-localized, i.e. centered on two different sites of the lattice, thus giving rise to much larger correlations than in the case of diagonal disorder. There is a vast literature in this respect which the authors might want to read:

C. Dasgupta, S.-K. Ma, Phys. Rev. B 22, 1305 (1980), where the strong-disorder renormalization group scheme is introduced.

D.S. Fisher, Phys. Rev. B 51, 6411 (1995); D.S. Fisher, A.Young, Phys. Rev. B 58, 9131 (1998). Shows that average correlations decay in a power-law fashion in the random Ising model in a transverse field. The infinite randomness fixed point is defined, which may be relevant to the authors.

G. Ramírez et al, J. Stat. Mech. P07003 (2014), where it is applied to the local XX model seen as a free-fermionic chain, discussing some subtle points, like the sign alternation of the effective couplings due to the fermionic character.

5.- The fluctuations in the particle number are a proxy to the entanglement entropy,

I. Klich, Journal of Physics A: Mathematical and General 39, L85 (2006).

and the entanglement entropy of different disorder systems has been analysed in detail in many works. For example,

G. Refael, J.E. Moore, Phys. Rev. Lett. 93, 260602 (2004); N. Laflorencie, Phys. Rev. B 72, 140408(R) (2005).

in summary, in the strong-disorder allows to compute entropies (and particle fluctuations) by counting the number of bonds which are cut if you separate a box from the environment.

7.- Regarding the efficiency of the transfer using the Raman-like passage, I find it very interesting. I still have some questions. One of them is about the Lieb-Robinson (LR) bound: you can not propagate an excitation through a quantum system faster than what is fixed by the LR bound. In a homogeneous system, the minimal time of arrival of any perturbation will be proportional to the system size. Thus, I do not find this surprising.

8.- In the same topic, why do the authors obtain a better transfer probability in the disordered case? Do the single-body wavefunctions somehow localize near the edges? Or perhaps it is merely an effect of the reduction of the LR time?

Some minor things:

9.- In they introduction, the authors mention "many brunches".

10.- The general behavior of long-ranged spin chains, and the critical values of the exponents at the end of the first paragraph deserve some citation.

11.- Line 84, the authors seem to imply that correlation between long distance sites of the chain will imply the possibility to immediately transfer information or energy between them. Obviously, this is not the case, and I am confident that the authors are perfectly aware of this. They should rewrite the explanation to account for this.

13.- The observable chosen to find out about the localization of the modes, the fluctuations of the number of particles in the left half of the chain, Eq. (3), does not seem to be very suitable for me. Most bilocalized states will not stretch across the boundary between both halves of the chain. The authors should justify their choice better.

14.- Line 149, "error propagation" sounds like a first year student. Uncertainty propagation sounds more professional.

15.- Is Fig. 3 showing the correlation between both extremes of the chain? This value can be analytically obtained for a homogenous chain.

16.- The authors should make it more clear when they are using diagonal/off-diagonal/no disorder, and when they are using nearest-neighbors or long-ranged hoppings. Sometimes it is not easy to find out.

Report

This article discusses protocols to transfer an excitation across an
XX spin-chain with disorder, comparing their efficiency with that of
an homogeneous counterpart. The topic is rather interesting and
timely, the methodology is probably appropriate and the results seem
correct. Yet, I have several important questions about the methodology
and the analysis of results. Furthermore, I guess that the authors
have missed important items in the literature which might help explain
their results. Therefore, I am willing to recommend this article for
publication only if they can successfully explain and/or correct these
issues.

Requested changes

Cited in the shortcomings section.

  • validity: good
  • significance: good
  • originality: good
  • clarity: low
  • formatting: good
  • grammar: excellent

Author:  Nikolaos Palaiodimopoulos  on 2022-11-04  [id 2984]

(in reply to Report 1 on 2022-10-13)

Response to the Report

Report

This article discusses protocols to transfer an excitation across an XX spin-chain with disorder, comparing their efficiency with that of an homogeneous counterpart. The topic is rather interesting and timely, the methodology is probably appropriate and the results seem correct. Yet, I have several important questions about the methodology and the analysis of results. Furthermore, I guess that the authors have missed important items in the literature which might help explain their results. Therefore, I am willing to recommend this article for publication only if they can successfully explain and/or correct these issues.

We are grateful to the Referee for carefully reading our manuscript and providing well justified comments and criticism. Some of these comments stem from our imprecise statements and formulations, which we have now corrected.

Below we give a point-by-point response to all the questions and comments.

Strengths

1.- The topic is very interesting and timely.

2.- The analysis of excitation transfer in disordered chains seems to pose very interesting questions.

We thank the Referee for the overall positive evaluation of our work.

Weaknesses

1.- First of all, the authors claim that the XX model is "interacting" in the abstract itself. Yet, all their discussion is done in terms of single-particle modes. These claims are not compatible. Of course, I guess that they are assuming that the XX model has been re-written as a free fermionic Hamiltonian using the Jordan-Wigner transformation. Moreover, I have several crucial issues in this respect. First of all, the authors should make their claims explicit. If they start out with an XX model and convert it to a free-fermionic model, they should say so.

What we mean is the exchange interactions between the spins, and more specifically resonant dipole-dipole interactions that have an 1/r^3 dependence on the distance. The operators σ^- and σ^+ in the Hamiltonian (1) are the spin raising and lowering operators, and this XX Hamiltonian does not contain the interaction terms of the (Ising) type σ^z σ^z. Such dipole-dipole exchange interactions are present in many physical realizations of the XX spin chains, e.g., Frenkel excitons, photosynthetic complexes, NMR in large molecules, chains of Rydberg atoms, etc. Hamiltonian (1) is isomorphic to the Hubbard Hamiltonian for spinless fermions or hard-core bosons with long-range hopping, and we focus on the single-excitation (single particle) dynamics of the system.

In the abstract and throughout the manuscript, we now always specify "exchange interaction" or "hopping" to avoid confusion.

2.- The long-range XX model can not be converted into a long-range quadratic free-fermionic Hamiltonian through the Jordan-Wigner transformation. The reason is the non-locality of the fermionic operators, which only cancels out in the 1D case.

We fully agree. Unlike other studies considering spin-chains with nearest-neighbor exchange interactions, we do not assume fermionization via the Jordan-Wigner transformation, since for a system with long-range excitation hopping this would lead to the appearance of quartic terms in the Hamiltonian, as the Referee points out.

3.- Even if we were using the short-ranged XX model, I still do not have a clear idea of how they build the quantum states. Indeed, the ground state of the XX model can be formed in this case as a Fermi sea (or Dirac vacuum, if you prefer). Perhaps they are focusing on the single-particle sector, i.e. nearly-maximal magnetization in the XX-model. If so, they should explain it, but I expect this is not the case, because the results would be far less interesting.

We indeed focus on the dynamics of single excitations in long-range hopping and disordered spin chains, as we state several times in the manuscript. But we contend that it is still interesting and relevant for many physical realizations of the spin chains and their use for excitation or polarization transfer. Long-range hopping on a disordered lattice is a non-trivial and interesting problem even for a single particle case, but we should emphasize that we do not study many-body localization or other interaction effects.

We clearly formulate the problem in Sec. 2, where we now explicitly say that the spin chain is assumed initially fully polarized, with all the spins unexcited. We need this assumption to exclude the interactions between the excitations (particles), since our main goal is to understand the excitation (particle) localization or delocalization in disordered lattices and their use for excitation transfer.

4.- The authors base their analysis of the wavefunctions assuming that they all fit the exponential ansatz, eq. (2). Yet, this is known not to be the case for some of the studied cases. Indeed, for local off-diagonal disorder, the single-body modes are known to be bi-localized, i.e. centered on two different sites of the lattice, thus giving rise to much larger correlations than in the case of diagonal disorder. There is a vast literature in this respect which the authors might want to read:

C. Dasgupta, S.-K. Ma, Phys. Rev. B 22, 1305 (1980), where the strong-disorder renormalization group scheme is introduced.

D.S. Fisher, Phys. Rev. B 51, 6411 (1995); D.S. Fisher, A.Young, Phys. Rev. B 58, 9131 (1998). Shows that average correlations decay in a power-law fashion in the random Ising model in a transverse field. The infinite randomness fixed point is defined, which may be relevant to the authors.

G. Ramírez et al, J. Stat. Mech. P07003 (2014), where it is applied to the local XX model seen as a free-fermionic chain, discussing some subtle points, like the sign alternation of the effective couplings due to the fermionic character.

We are of course aware that for, e.g., off-diagonal disorder and long-range hopping, the wavefunction is not exponentially localized around the (shifted) Dyson peak (E~-0.22 J) and is delocalized at the upper edge of the spectrum (E~2.4J), and we clearly say that in the manuscript. For such states, an exponential fit is not accurate. But our aim is to estimate the spatial extent of the wavefunction ξ which will determine the longest disordered spin chain that can still support efficient excitation transfer. The exponential ansatz is then adequate, and to verify it we also calculate the excitation number variance (equivalent to the entanglement entropy, as discussed below), which we plot in Fig. 2 next to the localization length. The two measures are fully consistent with nearly identical profiles, even when the wavefunction is completely delocalized, ξ ~ N. We average over many independent realizations of the chains to produce the graphs in Fig. 2.

Concerning the bi-localized states, we thank the referee for bringing this point up, and we now cite the two most relevant references [33,34]. Indeed, bi- and multi-localized eigenstates may appear for certain realizations of disorder. We were worried about such states, in the context of applicability of inverse participation ratio (IPR) [31] to quantify the localization. And we deduced that IPR is not suitable for our purposes [32], since it cannot determine whether a wavefunction is spatially localized on a number of neighboring sites or is delocalized on a similar number of remote sites.

The Referee is completely right in that if we have a bi- or multi-localized state and fit it with a smooth exponential ansatz, we would get a long localization length ξ of the order of the distance between the wavefunction peaks. But this is not wrong for our purposes, as we note after eq. (2), since we need ξ to deduce the length of the chain that can mediate excitation transfer, and such an extended wavefunction would be suitable for that purpose. Again, we compare the localization length deduced from the exponential fit with the excitation number variance; states with two or more peaks separated by large distances will yield large localization length and large number variance, while states strongly localized around one position will have small localization length and small number variance.

5.- The fluctuations in the particle number are a proxy to the entanglement entropy, I. Klich, Journal of Physics A: Mathematical and General 39, L85 (2006).

and the entanglement entropy of different disorder systems has been analysed in detail in many works. For example,

G. Refael, J.E. Moore, Phys. Rev. Lett. 93, 260602 (2004); N. Laflorencie, Phys. Rev. B 72, 140408(R) (2005).

in summary, in the strong-disorder allows to compute entropies (and particle fluctuations) by counting the number of bonds which are cut if you separate a box from the environment.

Indeed, for single-excitation (single-particle) states in a chain split into two parts, the number variance Δn^2 and the entanglement entropy S are equivalent, S >= (4 ln 2) Δn^2, since all entanglement stems simply from the excitation number fluctuations in that partition. In fact, in Fig. 2 we initially used the entanglement entropy to verify that our localization length correctly captures the extent of the single-particle wavefunction. But then we decided to use the particle number variance as a simpler and more experimentally relevant measure of localization. We have added a sentence noting the equivalence of S and Δn^2 and citing the relevant papers [35,36].

7.- Regarding the efficiency of the transfer using the Raman-like passage, I find it very interesting. I still have some questions. One of them is about the Lieb-Robinson (LR) bound: you can not propagate an excitation through a quantum system faster than what is fixed by the LR bound. In a homogeneous system, the minimal time of arrival of any perturbation will be proportional to the system size. Thus, I do not find this surprising.

We fully agree with the Referee's remark on the Lieb-Robinson bound. Even for power-law interacting systems 1/r^a, as long as a>= 2d+1 (with d being the system's dimension), the time needed for the propagation of information scales linearly with the size of the system [50-52]. But the adiabatic transfer protocol is much slower than what is allowed by the LR bound.

The linear scaling of the transfer time with the system size, Figs. 4 and 5, stems from our requirement of avoid leakage of the excitation to the undesired non-resonant states which restricts the boundary couplings to J_{s,r} ~ 1/ sqrt(N) leading to τ ~ N. But this scaling is consistent with the LR bound which is in fact a lower (and the lowest) bound on excitation transfer time, although with a reduced transfer efficiently due to the leakage. This applies to both static coupling and adiabatic coupling protocols, but of course the adiabatic transfer, while using the same extended resonant state of the chain (see below) is, by definition and construction, much slower.

We now mention the Lieb-Robinson bound and the relevant references in the discussion on linear scaling of the transfer time τ.

8.- In the same topic, why do the authors obtain a better transfer probability in the disordered case? Do the single-body wavefunctions somehow localize near the edges? Or perhaps it is merely an effect of the reduction of the LR time?

I sec. 3 we identify the single-excitation wavefunctions that are the most extended (not necessary localized at the edges) states to be used for the excitation transfer in finite spin chains of proper length (not exceeding the localization length, however it is defined). We find that the the most extended wavefunctions are around the shifted Dyson peak at energy E~-0.22 J and possibly at the upper edge of the band E~2.4 J. Then in sec. 4 we see that the states around upper edge, even though delocalized, have small support at the edges of finite chains, which means they only weakly couple to the input and the output spins. So in sec. 5 we focus on the state at energy E~-0.22 J and we resonantly couple the sender and receiver spins to that state. The system effectively becomes a three-state system: the excitation is at the sender spin, the selected state of the chain, and the receiver spin.

Now in a three-state system, one can just switch on the coupling and wait till the excitation travels from the initial state to the final state, which is what the static transfer protocol does. It works reasonably fast and the the transfer efficiently is limited. Because the system is not an ideal three-level system and the intermediate spin-chain eigenstate is not perfect: from realization to realization there are variations of its energy and localization length. But we know from AMO physics a more robust adiabatic protocol for population transfer in a three-state system: STIRAP, so we use it. It is more robust because it essentially avoids populating the intermediate state and works even when the intermediate state is not exactly resonant, as explained in Sec. 4 (see eq. (9) and thereafter).

There may be a misunderstanding concerning what is illustrated in the plots of Section 5. We suspect that the caption of Fig. 7 may be misleading, therefore we have slightly reformulated it. Both Fig. 6 and Fig. 7 depict the transfer probability versus the length of the chain in the presence of disorder. In Fig. 6 we show the performance of the static coupling protocol, comparing the long-range exchange interactions with the nearest neighbor case. In Fig. 7 on the other hand, we consider only chains with long-range exchange interactions, and we compare adiabatic passage protocol with the static coupling protocol.

Some minor things:

9.- In they introduction, the authors mention "many brunches".

We have corrected this typo.

10.- The general behavior of long-ranged spin chains, and the critical values of the exponents at the end of the first paragraph deserve some citation.

We have added the citations on the physical systems with the 1/r^3 (dipole-dipole) exchange interactions. At this point in the text we have not yet discussed disorder and critical behavior, which comes later in the text with proper citations.

11.- Line 84, the authors seem to imply that correlation between long distance sites of the chain will imply the possibility to immediately transfer information or energy between them. Obviously, this is not the case, and I am confident that the authors are perfectly aware of this. They should rewrite the explanation to account for this.

Of course we do not mean that an extended state can immediately transfer energy or information between distant locations. What we want to say is that we need to find appropriate states (both extended and strongly coupled to the sender and receiver spins) that can mediate such a transfer. We have reformulated the text to avoid confusing the reader.

13.- The observable chosen to find out about the localization of the modes, the fluctuations of the number of particles in the left half of the chain, Eq. (3), does not seem to be very suitable for me. Most bilocalized states will not stretch across the boundary between both halves of the chain. The authors should justify their choice better.

For us, the main parameter to characterize the modes is the localization length ξ, which is large if a mode is delocalized or bi-localized, and small if the mode is localized around a few neighboring lattice sites. And this holds independently on whether or not we (mentally) divide the chain into two halves. As we say in the manuscript (e.g. lines 122-124, 172-175, 214-217), we use the excitation number variance in one half of the chain as a "sanity check" that our ξ captures correctly the localization properties of the system, and it does, as verified in Fig. 3. This verification is warranted since other colleagues and now the Referee voiced legitimate concerns that not all the states are exponentially localized while we fit them all with an exponent. But as we argued above and also state in the manuscript (lines 116-118), our aim is not a perfect fit of the wavefunction, but a characterization of its spatial extent, and ξ is then a good measure for that, as also verified by Δn^2 (or equivalently the entanglement entropy).

We mention the localization length in the abstract and many times in the manuscript as a critical parameter for our purpose.

14.- Line 149, "error propagation" sounds like a first year student. Uncertainty propagation sounds more professional.

We agree, "uncertainty propagation" sounds better, which we now use.

15.- Is Fig. 3 showing the correlation between both extremes of the chain? This value can be analytically obtained for a homogenous chain.

That is true: a simple analytical expression exists for chains with nearest-neighbor hopping, while for the longer-range hopping the analytical expression involves sums of the terms over the range m as in Eq. (6), which is cumbersome and not very revealing. We therefore do not present it. Instead, in Fig. 3 we show the correlations between the two ends of the chain vs the energy, since this figure clearly illustrates the physics behind our choice of the suitable energy range.

16.- The authors should make it more clear when they are using diagonal/off-diagonal/no disorder, and when they are using nearest-neighbors or long-ranged hoppings. Sometimes it is not easy to find out.

We have made modifications in the text to clearly say when necessary that we deal with nearest-neighbor or long-range hopping and in the absence or presence of what kind of disorder.

---

## Round 2 · Referee Report · Anonymous · 2022-11-24

Report

The authors have successfully cleared all my doubts and made the required modifications to the manuscript. I am glad to recommend it for publication in SciPost Physics Core.

Requested changes

None. :)

---

## Round 2 · Author Response

Dear Editor and the SciPost Team members,

Thank you very much for considering our manuscript and sending it to a reviewer.
We have carefully considered the report of a Referee and made the necessary
amendments in our manuscript. We present a detailed response to the Referee's
report, followed by the summary of changes.

Sincerely,
N. E. Palaiodimopoulos,
on behalf of all the authors

---

## Round 2 · List of Changes

In the abstract, and throughout the manuscript, we now always specify
"exchange interactions" to avoid the confusion.

The second paragraph of Sec. 2 is modified and expanded for clear formulation
of the problem.

In the last three lines of page 4, we added a new text stating the relation
between the number variance and entanglement entropy and citing refs. [35,36].

The first sentence in the second paragraph of Sec. 4 is modified for clarity.

In the paragraph "Static coupling to the chain" in sec. 4 we have added
new text on the linear scaling of the transfer time and its relation with
the Lieb-Robinson bound, citing refs. [50-52].

A number of corrections and improvement of the text and figure captions
were made.

New references [33,34,36,50,51,52] were added, other references appropriately
shifted.

You are currently on this page

Resubmission scipost_202209_00031v2 on 4 November 2022

---

## Editorial Decision

published